# Estimating the Burden of Alcohol on Ambulance Callouts through Development and Validation of an Algorithm Using Electronic Patient Records

**DOI:** 10.3390/ijerph18126363

**Published:** 2021-06-11

**Authors:** Francesco Manca, Jim Lewsey, Ryan Waterson, Sarah M. Kernaghan, David Fitzpatrick, Daniel Mackay, Colin Angus, Niamh Fitzgerald

**Affiliations:** 1Institute of Health and Wellbeing, University of Glasgow, Glasgow G12 8QQ, UK; daniel.mackay@glasgow.ac.uk; 2Business Intelligence Department, Scottish Ambulance Service, Edinburgh EH12 9EB, UK; ryan.waterson@nhs.scot (R.W.); sarahmichelle.kernaghan@nhs.scot (S.M.K.); 3Faculty of Health Sciences & Sport, University of Stirling, Stirling FK9 4LA, UK; david.fitzpatrick@stir.ac.uk (D.F.); niamh.fitzgerald@stir.ac.uk (N.F.); 4School of Health and Related Research, University of Sheffield, Sheffield S10 2TN, UK; c.r.angus@sheffield.ac.uk

**Keywords:** ambulance callouts, burden of alcohol, algorithm development, routine health records, paramedics, Scotland

## Abstract

Background: Alcohol consumption places a significant burden on emergency services, including ambulance services, which often represent patients’ first, and sometimes only, contact with health services. We aimed to (1) improve the assessment of this burden on ambulance services in Scotland using a low-cost and easy to implement algorithm to screen free-text in electronic patient record forms (ePRFs), and (2) present estimates on the burden of alcohol on ambulance callouts in Scotland. Methods: Two paramedics manually reviewed 5416 ePRFs to make a professional judgement of whether they were alcohol-related, establishing a gold standard for assessing our algorithm performance. They also extracted all words or phrases relating to alcohol. An automatic algorithm to identify alcohol-related callouts using free-text in EPRs was developed using these extracts. Results: Our algorithm had a specificity of 0.941 and a sensitivity of 0.996 in detecting alcohol-related callouts. Applying the algorithm to all callout records in Scotland in 2019, we identified 86,780 (16.2%) as alcohol-related. At weekends, this percentage was 18.5%. Conclusions: Alcohol-related callouts constitute a significant burden on the Scottish Ambulance Service. Our algorithm is significantly more sensitive than previous methods used to identify alcohol-related ambulance callouts. This approach and the resulting data have potential for the evaluation of alcohol policy interventions as well as for conducting wider epidemiological research.

## 1. Introduction

Alcohol constitutes a significant burden on emergency services in the UK [1], with the potential to undermine or delay emergency service provision to other incidents. This is particularly true for ambulance services, which often represent patients’ first, and sometimes only, contact with health services. Acute alcohol episodes contribute to making ambulance clinicians’ work more difficult and risky: in 2010, more than 50% of ambulance staff who responded to a UK survey reported experiencing injuries or sexual harassment whilst dealing with drunk members of the public [1]. Despite this, the burden of alcohol on ambulance services has not been extensively researched and is not routinely monitored.

In the UK, previous studies have found varying levels of burden on ambulance services arising from alcohol consumption. In 2013/14, the London Ambulance Service reported from an audit of emergency calls that 6% of calls for an emergency ambulance were associated with alcohol [2]. In the North East of England, researchers used manual examination of ambulance patient records to estimate that alcohol incidents accounted for 10% of total callouts [3], while in Scotland in 2015 respondents to a survey of ambulance clinicians estimated that 17% of incidents on weekdays and 42% on weekend nights involved alcohol [4]. It is unlikely that these differences can be fully explained by geography but are more likely to arise from the different methods used to assess whether alcohol was involved, and the different data sources used (calls, staff surveys and callouts). The specificity and sensitivity of these different methods are not frequently assessed, meaning these estimates could be biased and lead to inconsistent or misleading comparisons across regions. The North East of England study found that the reading of free-text in individual patient records allowed the identification of approximately three times the number of alcohol-related ambulance callouts, but this was labour intensive and was only performed for a sample of records [3].

Outside of the UK, the National Ambulance Surveillance System (NASS) has been established in Australia to use ambulance data for the surveillance of a wide range of presentations [5,6,7]. Lubman et al. (2020) describe NASS as a complex and coordinated system employing 23 researchers to monitor and map acute harms related to alcohol and other drug consumption, using ambulance callout data from services covering 82.5% of the Australian population. This system, in which researchers read and code full patient records, overcomes the challenges identified in the UK and allows for effective monitoring, description and mapping of ambulance callouts related to alcohol consumption [8] and substance misuse [9]. Such information is valuable for policymaking, including policy development and evaluation, but is resource-intensive to gather in this way.

Data from ambulance records have been used as a complementary source of information to traditional surveillance systems for other illnesses, including respiratory infections [10,11,12]. However, there is no general agreement on how to use this information for surveillance purposes as the type of ambulance data used and analysed (assessment from patients’ initial calls [10], dispatch data, callout data, or paramedic surveys [13]) once again varies across studies.

The aim of this study was to improve the assessment of the burden of alcohol on the ambulance service. To attain this, we developed a low-cost and easy to implement automatic algorithm to screen free-text records in electronic patient record forms from ambulance callouts. In this paper, we both describe how the algorithm was developed and validated and present estimates on the burden of alcohol on ambulance callouts in Scotland between 2016 and 2019.

## 2. Materials and Methods

Below, Section 2.1, Section 2.2, Section 2.3 and Section 2.4 focus on how the algorithm was developed and assessed. We briefly describe in Section 2.5 how the algorithm was then used to estimate alcohol-related ambulance callouts in Scotland between 2016 and 2019.

### 2.1. Study Setting and Dataset

The Scottish Ambulance Service (SAS) is a population-wide service and part of the National Health Service in Scotland, free at the point of delivery. SAS serves a population of 5.5 million people and attends more than half a million incidents annually. Alcohol-related callouts can currently be recorded in two main ways by ambulance clinicians when completing electronic patient record forms (ePRFs) on tablet devices at the scene of the incident. Clinicians can either select an on-screen alcohol “flag” to indicate that alcohol was a contributing factor in the callout and/or describe how alcohol was a factor in free-text fields in the ePRF. The “flag” is what currently has been used from SAS to determine whether a callout involved alcohol. Personal communication with ambulance staff explained that paramedics in practice may sometimes not use the alcohol flag on the ePRF, rather alcohol involvement would be recorded in a free-text report completed after attending to the patient. Therefore, the alcohol flag was deemed likely to underestimate the burden of alcohol on the ambulance service. Furthermore, the likelihood of clinicians selecting the alcohol flag depends in part on the prominence of that flag, which has changed in different versions of the SAS ePRF system. This makes it difficult to use the flag alone for examining trends in alcohol-related callouts or the impact of alcohol policy changes that might influence such trends.

We have automated the process of reviewing free-text fields in ePRFs by building an algorithm capable of classifying ambulance callouts as alcohol-related, using the information recorded by ambulance clinicians in free-text fields. Our development of the algorithm was based on a sample of SAS ePRFs deliberately selected to include around 1000 alcohol-related callouts, extracted between 2015 and 2019. Earlier audits suggested that around 10% of total callouts would be alcohol-related; however, in order to reduce the number of records manually scrutinised by the paramedic, we sampled more callouts coming from periods when the number of alcohol-related incidents was likely to be higher (i.e., weekend night-times—from 6 p.m. to 6 a.m.). Using a manual audit of a small sample of full ePRFs at these times, we estimated around 27% of callouts at these times were related to alcohol. We therefore sampled 3600 callouts from these weekend night-times, a further 1200 from the rest of the week, and 616 additional records sampled at random from across the entire week, in case we would have found a number lower than 1000 callouts identified as alcohol-related. This gave us a total sample of 5416 callouts.

### 2.2. Assessment of Callouts as Alcohol-Related

Every ePRF contains tick box sections describing clinical and presentation characteristics of the patient as well as several open free-text fields where paramedics enter a description of the context and circumstances of the callout using their own words. As no gold standard exists to classify alcohol-related callouts, an experienced paramedic (SK) interrogated the sample of free-text from ePRFs and used her professional judgement to classify the callout as alcohol-related or not. We defined “alcohol-related callout” as any callout where alcohol had been recorded on the ePRF as a primary cause for care (i.e., alcohol intoxication or alcohol dependence) or in those calls where the consumption of alcohol was recorded in association with the presenting condition/injury. Examples of the latter are calls relating to mental health crises, falls or assaults and consumption of alcohol was a consideration in ongoing patient assessment, treatment and care. All uncertain entries were then cross checked by a second experienced paramedic (DF) and resolved by consensus.

This classification was considered the “gold standard” for judgement of whether a callout was alcohol-related or not. Due to information governance concerns, researchers could not have access to the full free-text of every ePRF, but only paramedics or SAS staff could view and analyse them. Therefore, the paramedic also recorded verbatim, including any misspellings, any phrases in the free-text entered in each ePRF which (i) were used to classify the callout as alcohol-related, or (ii) might result in incorrect classification as alcohol-related using an automated algorithm (i.e., text containing common alcohol terms, but where the overall callout was not judged to be alcohol-related, referred to hereafter as “misleading terms”). The classification for all sampled records and extracted phrases where relevant were recorded in a spreadsheet. Researchers worked with this restricted dataset.

### 2.3. Algorithm Development

The dataset, consisting of the classified patient records and free-text extracts, was divided into a training set (80%, 4327 records) and a validation set (20%, 1089 records). Validation and training sets were not split randomly, as a non-random sample is preferable for internal validation purposes [14]. We used two different approaches to algorithm development: “manual” and “machine learning” (ML). The development and operation of the manual algorithm for identifying alcohol-related callouts can be summarised in five stages:(a)Cleaning the extracted sections of text in the dataset for both alcohol-related and misleading terms (e.g., removing extra spaces, removing punctuation and excluding the “stop words”).(b)Identifying words common to the callouts classified by alcohol-related based on their frequency (recurrence in more than 2.5% of alcohol-related callouts) and expert opinion (e.g., extra words identifying unambiguously alcohol-related callouts such as names of specific beverages appearing in some of the remaining records classified as alcohol-related).(c)Looking at the recurrence of words identified in (b) within the misleading terms. Focusing on the combination of one word before and one word after the words in (b) within the misleading terms. Identifying the most frequent combinations.(d)Identifying and correcting the most common spelling errors in ePRFs of words identified in (b) and (c).(e)Identifying every callout as “alcohol-related” whenever there was at least one of the “alcohol-related terms”, except those excluded by the combinations in point (c).

In step (a), some of the words were reduced to their stem version to also include their declension, and others were maintained in their entire original form as differences could have been meaningful for the specific disease context (e.g., drink, drinks, drunk, drank could have different connotations and the difference could be relevant when related to alcohol) (see Table A2 in Appendix A). In applying the manual algorithm to all SAS callout records, a record was deemed to be “alcohol-related” if it either was identified as alcohol-related using algorithm search of the free-text as above, or if the alcohol flag was selected on the ePRF by the ambulance staff.

The second approach used an ML algorithm based on a random decision forests [15] process and was developed using the same dataset as for the manual algorithm based on sections of free-text. Specifically, the alcohol-related words plus the alcohol flag were used as nodes of a random forest. Random forests are a series of algorithms which learn by the way an observation was classified (alcohol-related or not) and other characteristics (the free-text) to predict new observations through building a multitude of decision trees, one hundred in our case.

### 2.4. Assesment of Algorithm Performances

The algorithms were developed and their performance analysed in Stata version 16 [16]. The algorithms were assessed based on sensitivity, specificity and accuracy parameter estimates in the validation dataset. Sensitivity (Equation (1)) is the percentage of “true” alcohol-related callouts detected by the algorithm. It can also be interpreted as the probability that each algorithm will detect an alcohol-related callout, when the callout is judged to be alcohol-related using the gold standard of paramedic assessment.
(1)$$\frac{True\;positive}{True\;positive+False\;negative}$$

Specificity (Equation (2)) is the percentage of “true” non-alcohol-related callouts detected by the algorithm. It can also be interpreted as the probability that the algorithm detects a callout as non-alcohol-related, when the callout is judged not to be alcohol-related using the gold standard.
(2)$$\frac{True\;negative}{True\;negative+False\;positive}$$

Accuracy (Equation (3)) is a measure of statistical bias. It can also be interpreted as the proximity of measurement results to the true value [17].
(3)$$\frac{True\;positive+True\;negative}{Total\;negative+Total\;positive}$$

Regarding the manual algorithm, different selections of words were tried and their performances were assessed. We selected the combination of words providing the best performance. The selected alcohol-related words (step (b) above) were the following: alcohol, drink, intoxication, vodka, bottle, drunk, buckfast, whisky, cider, beer, gin.

Table 1. Combination of words to remove from alcohol-related terms.shows the combination of words to remove (step (c)). Table A1 and Table A2 describe the list of selected stop words (step (a)) and the main root words, variations and spelling mistakes we included (step (d)).

### 2.5. Algorithm Application to Full SAS Dataset

The final selected algorithm based on the 5416 extracted records was applied by SAS analysts in SQL language to all ePRF records with complete free-text in the SAS data warehouse. Given its performance results (see below) and the ease to be implemented within the SAS warehouses, the manual algorithm was chosen for the estimation of the burden of alcohol on the Scottish Ambulance Service. The algorithm extracted monthly data for callouts deemed to be alcohol-related from 2016–2019 including the postcode (at district level) of the callout, callout characteristics (i.e., time of callout, emergency code, etc.) and patients’ demographics/characteristics. Total callout data (including callouts judged to be non-alcohol-related) were also obtained. Descriptive statistics and graphs were prepared to provide estimates of the burden of alcohol on the ambulance service with particular focus on 2019, the most recent available year in the dataset.

## 3. Results

Below, we first describe the performance of our algorithms (Section 3.1) and then focus on the estimates of the burden of alcohol for the Scottish Ambulance System (Section 3.2).

### 3.1. Algorithm Performance

Results are presented on the complete free-text records applied by SAS analysts on the validation dataset. The validation dataset had a similar overall percentage of callouts determined to be alcohol-related by the paramedic (17.5%) to that detected by the algorithm when applied to the overall dataset (validation plus training) (18.5%). The manual algorithm performance was comparable with the ML algorithm, with differences only in third decimal digits in all the three indicators (see Table 2). The alcohol flag alone outperforms both the algorithms in terms of specificity but had much poorer sensitivity and overall accuracy. The current alcohol flag does not identify many false positives but the rate of true positives identified is less than half.

### 3.2. Alcohol-Related Callouts

The algorithm detected an increasing trend of alcohol-related callouts over the period of 2016–2019 as well as a clear difference in the volume of callouts between weekend days (Friday–Sunday) and weekdays (Monday–Thursday) (Figure 1). In 2016, the daily averages on weekdays and weekends were 161 and 243 alcohol-related callouts, respectively, whereas in 2019 they were 202 and 284. This translates into an increase between 2016 and 2019 of 25% and 14% for weekdays and weekends, respectively. The callouts show a seasonal pattern with two periods where the overall volume of callouts increased. The first period was during summer (from May to August, with the peak in July), then they rapidly decrease in September and they slightly increase every beginning of December until the 1st of January of the year after (creating a second peak) (Figure 1). It is of note that the 1st of January has on average 200 additional callouts compared to any other day of the year.

In 2019, there were 536,536 ambulance callouts, of which 86,780 (16.2%) were identified as being alcohol-related (Table 3). During weekends, alcohol-related callouts represented 18.5% of the total, whereas during weekdays the corresponding figure is 14.2%. The distribution of alcohol-related callouts was slightly different between weekends and weekdays. More than 20% of alcohol-related callouts were involving individuals in the 40–54 years age group. There was also variation by sex, with males representing more than 60% of alcohol-related callouts but less than 50% of non-alcohol-related callouts. The distribution of alcohol-related callouts over hours of the day differed across age groups: overall, more callouts were to individuals residing in areas of highest socio-economic deprivation than in other areas, and this was also true for alcohol-related callouts. Almost 1 in 5 callouts to those residing in areas of highest socio-economic deprivation were alcohol-related compared to 1 in 10 to those residing in areas of lowest socio-economic deprivation. Geographical variations were also present; alcohol-related callouts are more concentrated in urban areas, and their relative weight on the total number of callouts is greater in urban areas compared to rural. Alcohol-related callouts were more likely to be rated as serious than non-alcohol-related callouts, with the distribution of alcohol-related callouts oriented towards the two most severe emergency codes (red and purple are 21.4% of the alcohol-related ambulance callouts compared to 16.7% of non-alcohol-related ambulance callouts). Additionally, the distribution of callouts over the hours of the day differed across the day of the week and age group. Specifically, during weekend nights (6 p.m.–6 a.m.) the percentage of alcohol-related callouts was 28%, while in the rest of the nights, it was 19.5%. In particular, the percentage of alcohol-related callouts on weekends peaked between the hours of 9 p.m. and 1 a.m., whereas the peak 4 h period for weekdays was between 6 p.m. and 10 p.m. (Figure 2). Younger individuals’ callouts related to alcohol were more concentrated late in the night (more than 30% of callouts between 11 p.m. and 2 a.m. for 0–24 years age group) compared to older people (more than 30% of callouts between 5 p.m. and 9 p.m. for over 70 years age group) (Figure 3).

## 4. Discussion

### 4.1. Alcohol-Related Ambulance Callouts

Using a robust methodology that we developed and validated, we have identified a high burden of alcohol on ambulance callouts in Scotland; in 2019, we estimate that approximately 16 out of every 100 callouts were alcohol-related. In addition, the proportion of alcohol-related callouts during weekends was higher than during weekdays. Beyond weekends, seasonal trends with peaks in callouts corresponded with the months of December (in particular Christmas and new year holidays) and the months with greater hours of daylight in Scotland (May–August). Thirty-five percent of callouts classed as the most severe (red and purple) were identified as alcohol-related callouts. Given the average cost of an ambulance callout in 2019 [18], the total cost of alcohol-related callouts can be estimated at approximately GBP 31.5 million, though this figure would depend on the complexity of these calls compared to non-alcohol-related callouts.

These figures provide a robust estimate of the burden of alcohol on the ambulance service in Scotland and have direct policy relevance. These data could inform wider alcohol policy decisions at both local and national government level aimed at reducing this burden, including efforts to reduce alcohol problems and dependence, as well as addressing the peak of callouts occurring at weekends. These findings raise questions about the balance of risks and benefits of current alcohol consumption and harms (and related regulations), and whether further action is needed, particularly in the light of attempts to protect health service availability in times of capacity constraints and during extraordinary events such as pandemics or disasters. It is likely that the burden of alcohol on society could be reduced by interventions focusing on its availability, affordability or attractiveness [19,20], though it remains to be seen which interventions would most directly reduce alcohol-related ambulance callouts. Further analysis would be beneficial to better understand the increase in burden at the weekends, including the extent to which this is driven by disorder nearby to or relating to consumption in licensed premises and by alcohol consumption in homes. Qualitative work with paramedics is underway to better understand their experiences of alcohol-related callouts, including the relative contribution of acute alcohol intoxication versus chronic alcohol consumption and dependence.

### 4.2. Strengths and Limitations of the Algorithm

The algorithms we developed outperformed an existing electronic alcohol flag in terms of overall accuracy and sensitivity, enabling the identification of a much larger burden of alcohol-related callouts than previously reported in the Scottish Parliament [21]. A feature of our methodological approach was to embed the existing alcohol flag into the algorithms. This, in our view, makes intuitive sense—the paramedic at the scene was compelled enough to press this flag as they considered alcohol to be a contributing factor and we believe that it is unlikely that this would be pressed in error. Further, embedding this into our algorithm allowed the capture of any alcohol-related callouts with no explicit mention of “alcohol” in the free-text.

The algorithm described here overcomes many of the issues around sensitivity or self-selection biases described in similar studies which used ambulance dispatch data alone [11] or in combination with paramedics’ reports [13]. Furthermore, it is likely to be a very cost-effective approach compared to other systems such as the Australian system described above, in which multiple staff coded individual records. The development costs included time for a paramedic reading and screening 5416 ePRF records to classify them as alcohol-related or not, data governance management time, and researcher time to develop the approach to sampling, design and testing of the algorithm. While the limited involvement of paramedics due to disclosure reasons to screen ePRFs (one for a full assessment and another one to confirm doubtful observations) contained costs, this could have potentially reduced the quality of assessing a few records, as a higher number of reviewers to cross-examine records is usually preferred. The application of the algorithm to the SAS dataset required additional analyst and developer time. This would be the main ongoing cost of generating continuous data on the burden of alcohol on SAS, apart from costs of periodic refinement of the algorithm. Therefore, we believe that one of the main strengths of this method, beyond the high precision of the estimates, is the ease of implementation and the economical approach.

It is important to note that the results of our algorithms are based on sections of free-text filtered by a paramedic due to disclosure restrictions. Moreover, we excluded from our algorithms the sections of text coming from uncertain ePRF recording.

Whilst ML algorithms have been shown to be effective approaches for similar problems, due to their ability to identify underlying correlation structures, we did not find that our ML algorithm outperformed the manual algorithm. One explanation for this may be due to the restricted volume of free-text which was available. Indeed, the preferred way to build a random forest would have been directly on the original full free-text of the ePRF instead of on the sections of text selected through a manual interrogation of records. However, as researchers only had access to sections of free-text, with relatively low computational power, the preferred solution was not feasible.

Disclosure constraints were the main source of limitations of our algorithms. We outline the two main consequences of these limitations below. Firstly, the algorithm could under-fit the data because the word selection to build the algorithm was based on sections of free-text and not the full text contained in the ePRF. This could have generated a bias. As the algorithm is a further selection of keywords already selected and screened by a paramedic, it could not have adequately captured the underlying structure of the free-text, generating a bias due to an under-fitting of the algorithm in the training sample. Secondly, there could be a likely upward bias in the algorithm specificity. Particularly, the algorithm has been built on a population which contained more alcohol-related callouts than the average numbers in the population to gather more information on keywords related to alcohol, in an attempt to increase the proportion of true positives and therefore the sensitivity. However, the artificial increase in sensitivity could imply an indirect decrease in specificity. Specifically, artificially increasing the number of positives in our sample decreases the proportion of false negatives, i.e., decreasing the overall number of negatives, you also decrease the number of false positives, generating an upward bias in specificity (this could explain the high values in specificity in Table 1). These two causes of likely bias were driven by the limitation of access to the dataset for information governance reasons.

Furthermore, ambulance callouts to individuals with conditions where chronic alcohol consumption may have been a contributory factor, including cancers or heart attacks, are generally less likely to be classified as alcohol-related as the contribution of alcohol would normally not be observable to the paramedic at the scene in the way that acute alcohol consumption is. As a result, the figures presented here could underestimate the full burden of alcohol on the ambulance service.

It is worth noting that many of the terms used in the algorithm may be context/country-specific; indeed, a few terms such as beverage brands or slang can be common in certain countries and rare in others (e.g., “buckfast”, which is one of the selected words of the algorithm, is a low-cost caffeinated alcoholic drink popular in Scotland). For this reason, when developing similar algorithms in different contexts, appropriate terms should be selected not only based on their frequency in the ePRF, but they should also be validated by experts with relevant local experience (e.g., paramedics). Potentially, experts could also add other specific unambiguous terms to increase the sensitivity of the algorithm. Therefore, although we would expect that the implementation of a manual algorithm containing the same words selected above on a system based in a different country could achieve good results, it is likely to have worse performance than one tailored to the specific context. In addition, we believe that whereas the algorithm method would still be valid in Scotland over time, the selection of words could evolve as social habits (e.g., drinking patterns and popular drinks) change or new words or products come into existence. Although the frequency of words was found to be consistent over our time frame, we suggest that the selection of words and the algorithm performance should be regularly reviewed and checked to guarantee estimates to be sensitive and consistent over time.

### 4.3. Further Opportunities

We describe a method to undertake epidemiologic monitoring using paramedics’ notes in ePRFs, without requiring further work from paramedics or other ambulance clinicians. We describe the application of this system to alcohol-related ambulance callouts in Scotland. However, the same method could be implemented for other epidemiological investigations such as infectious disease outbreaks or chronic diseases, where free-text records may enable the identification of callouts of interest over and above any system “flag”. This approach not only allows the identification of relevant callouts but also the analysis of relationships between specific callout types and other social or natural factors with spatial and temporal dimensions (e.g., socioeconomic deprivation, traffic, pollution).

In this case, tracking changes in alcohol-related callout demand could help to assess how local behaviours have changed over time, or how certain public health policies or interventions could have affected alcohol consumption and related harms. Furthermore, monitoring how the concentration can change over the week and over the hours of the day could also help to plan the allocation of ambulance resources in times of capacity constraints. These data can also have the potential to inform local alcohol premises’ licensing policies and decisions, as is currently done with alcohol-related hospital admissions and death [22].

Finally, ambulance data have also been used to assist and complement the understanding of data in other fields. For instance, data from ambulance callouts linked with police data can provide additional information on security and violence in certain city districts. Specifically, previous studies found that between 66 and 90 per cent of ambulance incidents related to violence are not included in police data [23]; therefore, improved recording of alcohol in ambulance data could help to enhance our understanding of the scale of and spatio-temporal patterning of violence and alcohol-related violence and thus improve police and public health responses.

## 5. Conclusions

Between 2016 and 2019, the burden of alcohol for the Scottish Ambulance Service was high, with 86,780 alcohol-related callouts in 2019, representing 16.2% of total callouts. Further, the number of alcohol-related callouts increased between 2016 and 2019. Our methodological approach for identifying alcohol-related ambulance callouts is significantly more accurate than previous methods. This approach and the resulting data have the potential to evaluate alcohol policy interventions as well as for conducting wider epidemiological research.

## Figures and Tables

**Figure 1 ijerph-18-06363-f001:**
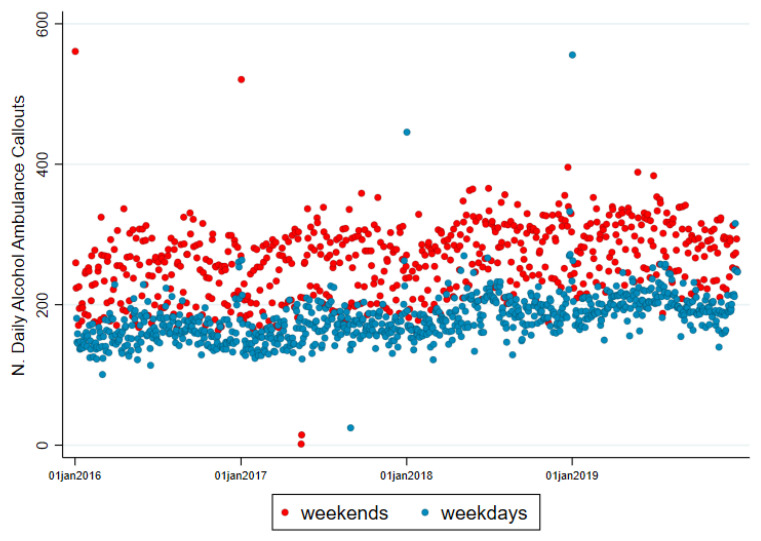
Daily alcohol–related ambulance callouts, 2016–2019.

**Figure 2 ijerph-18-06363-f002:**
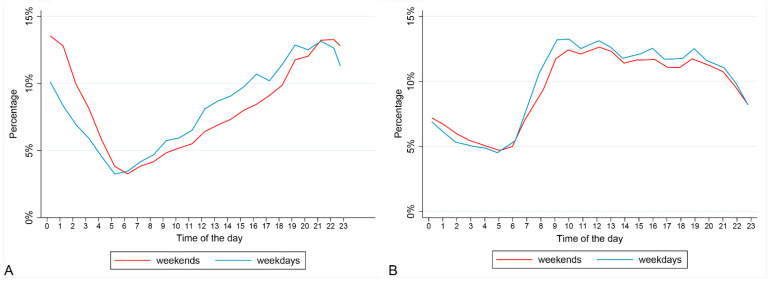
Distribution of alcohol-related callouts (panel **A**) and non–alcohol–related callouts (panel **B**) during hour of the day by weekend and weekdays.

**Figure 3 ijerph-18-06363-f003:**
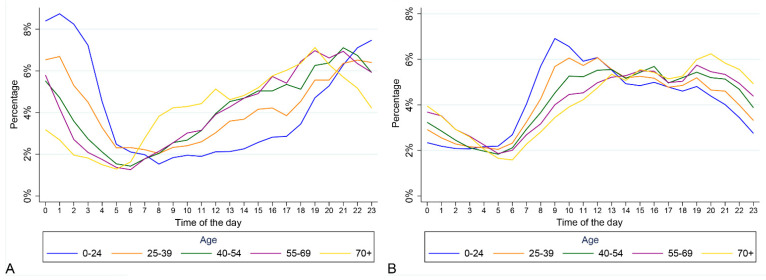
Distribution of alcohol-related callouts (panel **A**) and non–alcohol–related callouts (panel **B**) during hour of the day by age group.

**Table 1 ijerph-18-06363-t001:** Combination of words to remove from alcohol-related terms.

Main Word	Combination of Words to Exclude
Alcohol	“since alcohol” “any alcohol” “no alcohol” “or alcohol” “denies alcohol” “alcohol detox” “alcohol withdrawal”
Drink	“only drink” “any drink” “energy drink” “denies drink” “drink water” “not drink”
Intox	“appear intox” “not intox”
Bottle	“water bottle” “glass bottle”
Whisky	“one whisky”

**Table 2 ijerph-18-06363-t002:** Algorithm performance, including alcohol flag.

Statistic	Manual Algorithm	ML Algorithm	Alcohol Flag
Sensitivity	0.941	0.942	0.380
Specificity	0.996	0.996	1.000
Accuracy	0.986	0.987	0.890
Alcohol flag = current flag used by SAS to record alcohol-related callouts			

**Table 3 ijerph-18-06363-t003:** Descriptive statistics regarding alcohol-related and non-alcohol-related ambulance callouts in 2019.

	Alcohol-Related Callouts no. (%)	Non-Alcohol-Related Callouts no. (%)	Alcohol-Related Callouts as % of Total Callouts
Total	86,780	449,756	16.2%
*Day of the week*			
Sunday	15,663 (18.1)	65,447 (14.6)	19.3%
Monday	10,746 (12.4)	65,509 (14.6)	14.1%
Tuesday	10,657 (12.3)	63,925 (14.2)	14.3%
Wednesday	10,250 (11.8)	62,203 (13.8)	14.1%
Thursday	10,707 (12.3)	63,349 (14.1)	14.5%
Friday	12,526 (14.4)	63,806 (14.2)	16.4%
Saturday	16,231 (18.7)	65,517 (14.6)	19.9%
*Month of the year*			
January	7033 (8.1)	39,054 (8.7)	15.3%
February	6586 (7.6)	34,591 (7.7)	16.0%
March	7410 (8.5)	36,851 (8.2)	16.7%
April	7297 (8.4)	36,260 (8.1)	16.8%
May	7451 (8.6)	37,463 (8.3)	16.6%
June	7622 (8.8)	36,957 (8.2)	17.1%
July	7727 (8.9)	37,258 (8.3)	17.2%
August	7527 (8.7)	37,182 (8.3)	16.8%
September	7020 (8.1)	37,329 (8.3)	15.8%
October	6889 (7.9)	38,250 (8.5)	15.3%
November	6855 (7.9)	38,129 (8.5)	15.2%
December	7363 (8.5)	40,432 (9.0)	15.4%
*Emergency code* ^1^			
Green	147 (0.2)	965 (0.2)	13.2%
Yellow	48,250 (55.6)	242,937 (54.0)	16.6%
Amber	19,819 (22.8)	130,870 (29.1)	13.2%
Red	16,563 (19.1)	63,362 (14.1)	20.7%
Purple	1976 (2.3)	11,499 (2.6)	14.7%
Unknown	25 (0.03)	123 (0.03)	16.9%
*Age group (years)* ^2^			
0–24	12,758 (14.7)	41,298 (10.4)	23.6%
25–39	16,863 (19.4)	48,088 (12.1)	26.0%
40–54	19,632 (22.6)	55,252 (13.9)	26.2%
55–69	16,834 (19.4)	76,461 (19.2)	18.0%
70+	12,283 (14.2)	176,228 (44.4)	6.5%
*Sex* ^3^			
Female	31,612 (38.1)	218,471 (52.3)	12.6%
Male	51,378 (61.9)	199,634 (47.7)	20.4%
*Scottish Index of multiple deprivation decile for patient home address* ^4^			
1 (most deprived)	7284 (21.8)	29,836 (15.4)	19.6%
2	5246 (15.6)	25,681 (13.3)	17.0%
3	4554 (13.6)	24,504 (12.7)	15.7%
4	3919 (11.7)	21,655 (11.2)	15.3%
5	3111 (9.3)	19,167 (9.9)	14.0%
6	2491 (7.4)	17,684 (9.1)	12.4%
7	2241 (6.7)	16,236 (8.4)	12.1%
8	1699 (5.1)	14,310 (7.4)	10.6%
9	1591 (4.8)	13,094 (6.8)	10.8%
10 (least deprived)	1306 (3.9)	11,601 (6.0)	10.1%
*Scottish Index of multiple deprivation decile for callout location* ^5^			
1 (most deprived)	17,473 (20.1)	66,680 (15.0)	20.8%
2	12,568 (14.5)	57,671 (13.0)	17.9%
3	11,691 (13.5)	54,405 (12.2)	17.7%
4	10,102 (11.6)	49,029 (11.0)	17.1%
5	8715 (10.0)	45,223 (10.2)	16.2%
6	7355 (8.5)	44,378 (10.0)	14.2%
7	5795 (6.7)	37,472 (8.4)	13.4%
8	5286 (6.1)	34,927 (7.8)	13.2%
9	3695 (4.3)	29,427 (6.6)	11.2%
10 (least deprived)	3383 (3.9)	26,248 (5.9)	11.4%
*Rural/urban areas classified by callout location* ^6^			
Large urban area	36,107 (41.6)	159,817 (36.0)	18.4%
Other urban area	32,514 (37.5)	164,774 (37.2)	16.5%
Accessible small town	6154 (7.1)	36,425 (8.2)	14.5%
Remote small town	3046 (3.5)	17,292 (3.9)	15.0%
Accessible rural area	5360 (6.2)	42,732 (0.1)	11.1%
Remote rural area	2555 (2.9)	22,470 (0.05)	10.2%

^1^ Emergency codes are displayed in order of severity from green (least severe) to purple (most severe); ^2^ 8410 and 68,839 individuals did not have age recorded for alcohol-related and overall callouts, respectively; ^3^ 3772 and 31,601 individuals did not have sex recorded for alcohol-related and non-alcohol-related callouts, respectively; ^4^ 1, most deprived decile, 10 least deprived decile; 53,338 and 255,988 individuals did not have SIMD recorded for alcohol-related and non-alcohol-related callouts, respectively; ^5^ 717 and 4296 incidents did not have SIMD recorded for alcohol-related and overall callouts, respectively; ^6^ 1044 and 6246 individuals did not have rural/urban area recorded for alcohol-related and non-alcohol-related callouts, respectively.

## Data Availability

There is no public availability of these data for privacy reasons. Data was obtained from the Scottish Ambulance Service.

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
