# Peer review of "Estimating the Burden of Alcohol on Ambulance Callouts through Development and Validation of an Algorithm Using Electronic Patient Records"

_ijerph, 2021, doi:10.3390/ijerph18126363_

Round 1

Reviewer 1 Report

Overall, this is an interesting article especially since it is rare to see a combination of both manual diagnostic and machine learning (ML) to form a decision. However, there are some areas of improvement that are worth consideration. 

First, the paper appears to be somewhat dispersed as to what its ultimate focus is. Is it the improvement achieved in callout detection because of using a machine learning service or is it the status of alcohol-related call-out in the Scottish context? The reason this question arises is, the article in the abstract and the earlier of article have a good emphasis on the use of an ML algorithm in this alcohol call-out area - which is certainly valuable. However, when covering discussion and later part of the article, there appears to be an emphasis on the proportion of callouts and implications. Arguably, the emphasis can be in one of those focuses, like either suggesting the feasibility of ML algorithm-based approach which assists in better or comparable prediction than manual methods, perhaps in a faster way; or,  focusing explicitly on the proportion of alcohol callouts and the implications of thereof, with ML as a  supporting tool aiding the diagnostics. Potentially, the title suggests a preference for the first option.

Second, very early in the abstract, the paper introduces the term “free-text”, however, it is not until the later part of the article, like in Section 2.2, some idea of what free-text means is covered. Perhaps the authors may include a block diagram or figure of some sort and give an idea as to how ePRF looks like and what free-text implies within that. If I have understood proper, free-text basically means textboxes that allow open-ended responses by paramedics and health professionals, and which can therefore be an interesting avenue of exploration due to not being tied to a specific layout. If that is the case, may explain accordingly.

Regarding data collection, line 106, the authors refer to “oversample” – unclear as to what oversampling means in this case. Was it a duplication of ePRH data (oversampling has a particular meaning in the ML literature)? Or, was it that in the formed datasets more alcohol-related callouts were considered than the non-alcohol-related ones, especially so that the classification algorithm like random forest (RF) is not impacted? I am guessing it is the second one. Now, in that case, this looks more like an undersampling of the majority class. However, even in such a scenario, what was the proportion of alcohol and non-alcohol-related samples in training data? Did the validation set have the same proportion?

Regarding sampling, it is also not clear what the validation set and the referred “back-up” set imply. Typically, in ML literature, a validation set is used for tuning hyperparameters. However, it looks like the validation set used in this article was basically the test set. If that is the case may clarify so. Again, the term “validation” has a specific meaning in ML, and if the intended meaning is different, it is better to clarify. May further clarify the “back-up set”  - is it the out-of-sample set?

Following up on the above comment, what was/were the hyperparameter values for RF? How many trees were constructed? An ML approach is likely to vary considerably, in terms of performance, for various hyperparameters. Thus, more reflection concerning g this will avoid a reader’s confusion.

A unique aspect of this article is trying to compare manual undertaking with ML based undertaking. However, what is the implication of this comparison? Further the lines 185-205, where the paper refers to different manual algorithms based on different words, it is not clear what these different algorithms imply. Were any changes incorporated in the steps shown in lines 143-157? If not, then the algorithm is the same except for different parameter values – in this case, the words.

It is further not clear what "alcohol flag" means in Table 2 and later part of the article. It is to be noted that readers of this paper may stem from various disciplines with its multi-disciplinary nature. If such is the case, the authors may consider elaborating technical terms to cover for the diverse readers. 

Also, Table 2 shows the ML approach is comparable to the manual approach. Then, this raises the question as to what is the value of the ML in this domain then? I think the value comes from the automation and time saving the ML algorithm provides, and its comparableness to the manual approach is perhaps a strength - however, this may be clarified. 

Discussion is one place where the article can improve after fixing up its focus. The discussion is largely about the proportion of the alcohol-related cases, and potentially undermines the value of the paper which possibly can strengthen the focus on the use of ML as an interesting approach in the considered research domain. 

Author Response

We thank Reviewer 1 for the insightful comments that allowed us to improved our manuscript and we are pleased that our reasearch was found to be insightful and important. We addressed all comments in the PDF attachment

Reviewer 2 Report

Review of the manuscript “Estimating the Burden of Alcohol on Ambulance Callouts: 2 Development and Validation of an Algorithm using Electronic 3 Patient Records” submitted to the journal of IJERPH. In this work, the authors aimed to develop an algorithm to identify alcohol-related ambulance callouts in Scotland. This is meant to improve the assessment of alcohol-related cases using a cost-efficient and easy algorithm to screen free-text in ePRFs. Although this is a very important issue and the study has a point to make aside from the developed algorithm's usefulness, I find a few steps a bit concerning that would like to comment and receive the author's reply.

  • Having only 1 reviewer to review the forms and just checking it with a second one in case of not being sure is quite risky. In such cases, at least 2 independent evaluations for all the cases (if not three) and cross-examining the evaluation is preferred. Unless the case is confirmed by all the reviewers, the case either should be audited by another reviewer or discarded.
  • Although the study has a point and the analysis and algorithm show good performance, no clear explanation of how this algorithm is going to be used in real-time is provided.
  • L108-109: how many? It is not clear how many cases are considered here as a small dataset? 5,416?
  • Please provide the ML algorithm implemented in pseudocode

Author Response

We tahnk reviewer 2 for the insightful comments that imporve the quality of our manuscript. and we are pleased that our research was found to be important and interesting. All comments are addressed in the PDF attached

Round 2

Reviewer 2 Report

Based on the author's responses, seem there are lots of policies involved in terms of patient confidentiality and review process. Also they tried to address most of the comments here.